# Sharing information about domestic violence and abuse in healthcare: an analysis of English guidance and recommendations for good practice

Sandi Dheensa [ID] , Gene Feder [ID]

Domestic Violence and Abuse Health Research Group, Centre for Academic Primary Care, Population Health Sciences, Bristol Medical School, University of Bristol, Bristol, UK

**Correspondence to**
Dr Sandi Dheensa;
sandi.dheensa@bristol.ac.uk

## ABSTRACT

**Background** Over two million adults experience domestic violence and abuse (DVA) in England and Wales each year. Domestic homicide reviews often show that health services have frequent contact with victims and perpetrators, but healthcare professionals (HCPs) do not share information related to DVA across healthcare settings and with other agencies or services.
**Aim** We aimed to analyse and highlight the commonalities, inconsistencies, gaps and ambiguities in English guidance for HCPs around medical confidentiality, information sharing or DVA specifically.
**Setting** The English National Health Service.
**Design and method** We conducted a desk-based review, adopting the READ approach to document analysis. This approach is a method of qualitative health policy research and involves four steps for gathering, and extracting information from, documents. Its four steps are: (1) Ready your materials, (2) Extract data, (3) Analyse data and (4) Distill your findings. Documents were identified by searching websites of national bodies in England that guide and regulate clinical practice and by backwards citation-searching documents we identified initially.
**Results** We found 13 documents that guide practice. The documents provided guidance on (1) sharing information without consent, (2) sharing with or for multiagency risk assessment conferences (MARACs), (3) sharing for formal safeguarding and (4) sharing within the health service. Key findings were that guidance documents for HCPs emphasise that sharing information without consent can happen in only exceptional circumstances; documents are inconsistent, contradictory and ambiguous; and none of the documents, except one safeguarding guide, mention how coercive control can influence patients' free decisions.
**Conclusions** Guidance for HCPs on sharing information about DVA is numerous, inconsistent, ambiguous and lacking in detail, highlighting a need for coherent recommendations for cross-speciality clinical practice. Recommendations should reflect an understanding of the manifestations, dynamics and effects of DVA, particularly coercive control.

## STRENGTHS AND LIMITATIONS OF THIS STUDY

⇒ A strength of this study is that it is the first to review and analyse guidance for healthcare professionals around information sharing relating to domestic violence and abuse and has done so using robust methods.
⇒ Limitations include that this was not a systematic review.
⇒ Moreover, we were just two analysts, which may have limited the rigour of our analysis.

## INTRODUCTION

Domestic violence and abuse (DVA) is any incident, or pattern of incidents, of controlling, coercive, or threatening behaviour, violence, or abuse between people of any gender or sexuality aged 16 and above who are or have been intimate partners or family members.[1] Globally, 27% of ever-partnered women aged 15–49 have experienced physical and/or sexual violence from their intimate partner.[2] In England and Wales, an estimated 2.3 million adults aged 16–74 experienced DVA between March 2019 and 2020.[3] DVA damages mental and physical health[4] and is a public health and human rights issue. Coercive control, in particular, suppresses victim-survivors' autonomy, liberty, personhood and dignity.[5]

Analyses of hundreds of UK multiagency reviews of death and harm (domestic homicide reviews, safeguarding adults reviews, and serious case reviews)[6–14] show that the UK's National Health Service (NHS) has more contact with victims and perpetrators than any other agency or service.[7–10] One analysis illustrated that the NHS is the most common target for recommendations in domestic homicide reviews.[7]

A frequently cited failing across these analyses is that healthcare professionals (HCPs) did not properly document[6–12] and/or share information[6–15] related to DVA. Resultantly, no front-line professional had the whole picture of risk and no-one responded to the risk. The information in question included

risk factors or indicators for DVA (eg, relapse in mental ill-health or substance use, frequent emergency department attendance), perpetrators' threats to harm or kill, and explicit disclosures of DVA. The analyses showed inadequate sharing of information from the NHS to other agencies or services, as well between different parts of the NHS: namely between general practice, emergency departments, mental health, maternity and health visiting. Notably, failures to share within the NHS were more often related to information about the perpetrator, not the victim.[8] Several domestic homicide reviews cited a lack of communication about perpetrators between General Practitioners (GPs) and mental health services: for example, one involved a GP not alerting mental health services about a patient's non-adherence to medication, and another involved an acute hospital trust not telling a GP they had prescribed medication with potentially adverse mental health effects to a patient. In both cases, the patient went on to kill their female partner.[8] The analyses also highlighted that communication between mental health services and child and adult safeguarding was poor.[9] Moreover, reviews where victims had caring responsibilities for perpetrators (common in adult family homicides, eg, matricide cases) pointed out that professionals often excluded carers from care planning meetings where key information might have been shared and carer vulnerability recognised.[10] Finally, analyses highlighted insufficient referrals to, and attendance at, multiagency risk assessment conferences (MARACs)—where cross-sector professionals share information about high-risk DVA cases to devise coordinated action plans aiming to increase victim-survivor safety.[6–11 14] Analyses have emphasised a need to improve multiagency working to identify, assess and respond to risk.[6–9 11–14]

HCPs' reluctance to share information was partly because of confusion around sharing information without the patient's explicit consent.[9] Relatedly, some reviews indicated that HCPs decided not to share information about DVA because the patient had the capacity to withhold consent; in some cases, this was coupled with there being no formal safeguarding requirement to share information.[16 17] In the UK, formal safeguarding processes apply only when a person is 'vulnerable', that is, has care and support needs, is experiencing or is at risk of abuse or neglect (including DVA), and is unable to protect themself against the abuse or neglect, or the risk of it, because of the care and support needs in question.[18]

Consent is a lawful basis for sharing information[19] and it avoids mirroring the harmful dynamics between victim-survivors and perpetrators, where perpetrators wield power and control. However, sharing without consent is legitimate, for example, if it benefits victim-survivors and/or their children.[19] Different types of sharing require different types of consent, which might not always be obvious to HCPs: for example, referral to safeguarding or MARAC does not require consent or even victim-survivors' knowledge.[20 21] More generally, sharing health-related information within clinical teams,

which can include a wide range of staff who have not met the patient, can happen with implied consent.[22] Sharing information with NHS-based, but externally employed, DVA workers or agencies and third-sector mental health or substance services would usually rely on explicit consent.[22]

Research with primary and secondary care HCPs has echoed the findings from analyses of multiagency reviews: HCPs have reported being unsure about managing DVA information. HCPs reported that they did not consistently document DVA, were unsure about whether, where and how to document it, and how to do so safely.[23–29] In England, recommendations on DVA training for HCPs vary according to job role[30] and training does not explicitly address recording and sharing information. Most UK medical schools have poor DVA teaching provision[31] and junior doctors have reported feeling unprepared to manage DVA cases and confidentiality for patients brought into hospital by police.[32]

## Aim
To improve practice, we led a project called Recording and Sharing Information about DVA in Healthcare (RASDIH). It involved multiple research strands including an in-depth analysis of existing guidance about medical confidentiality, information sharing and DVA for HCPs (with a secondary focus on social care professionals) . We present this analysis here. We aimed to highlight commonalities, inconsistencies, gaps and ambiguities in guidance and in turn the tensions, dilemmas and complex ethical issues with which HCPs are faced when caring for patients affected by DVA. We chose not to include police since information sharing by police is largely guided by the Crime and Disorder Act rather than any specific focument. The findings from this analysis, as well as findings from the other research strands, informed recommendations for good practice in England. RASDIH was in turn part of a larger project called Health Pathfinder, which aimed to enhance the healthcare response to DVA. Reports from RASDIH[33] and Health Pathfinder are published elsewhere.[34]

## METHOD
We conducted a desk-based analysis of guidance documents, adopting the READ approach[35] to analyse the documents. This approach is a method of qualitative health policy research and involves four steps for gathering, and extracting information from, documents. Its four steps are: (1) Ready your materials, (2) Extract data, (3) Analyse data and (4) Distill your findings.

SD, a research fellow and applied social scientist, led on the analysis with support from GF, an academic GP. Having two different perspectives (clinical and non-clinical) was helpful for analysis, and differences in interpretation of documents were resolved through discussion.

### Ready your materials

Within the READ approach, researchers first establish parameters around the topic, where to search, and dates. We searched for documents that guided practice around medical confidentiality and information sharing, or DVA specifically. We decided to focus on the policies of national bodies from England that guide and regulate healthcare across specialities. With RASDIH's professional advisory group (comprising clinicians, medical confidentiality experts, 'DVA and health' researchers and DVA practitioners), we decided on the British Medical Association (BMA), General Medical Council (GMC), Nursing and Midwifery Council (NMC), National Institute of Health and Care Excellence (NICE), UK Caldicott Guardian Council and Department of Health and Social Care (DH) as key bodies. Our documents included reports, guidelines, position papers, recommendations and codes of practice published within the past 20 years. We checked each authoring body's website to ensure the document was the latest version. From our initial search results, we used backwards citation-searching to identify additional relevant documents from other national bodies in England, from the Social Care Institute for Excellence (SCIE), Adass and Local Government Association. We excluded speciality-specific documents (eg, from Royal Colleges) to give the review presented here boundaries, although the RASDIH full report contains a review of these documents[33].

### Extract your data

The READ approach does not prescribe any one way to extract data, but one straightforward way—which we adopted—is to use an Excel spreadsheet where each row represents a document, and each column represents a category of information. After closely reading each document in its entirety, we decided on key categories of information that we wanted to know from each document. These key categories include the headings shown in table 1 and what the documents stated about sharing information within the NHS and sharing from the NHS with other agencies or services. We then reread documents and extracted this information. Thus, our approach to data extraction was iterative and deductive, in that our categories of interest were predetermined. As we reread documents, we split these initial categories into subcategories. We then reread earlier documents again to ensure we had extracted all the relevant information. As we read, we also made notes about who wrote the documents, whether documents referenced each other, what they recommended on the same issues, whether recommendations had changed over time and the laws and ethical constructs underpinning the recommendations.

### Analyse data

Applying a specific analytical method once all data are extracted sheds light on what the documents state, overall, about the key topic, and on the commonalities and contradictions between documents. To achieve this

insight, the READ approach can be used in conjunction with different qualitative methodologies. We used the Framework Method[36] because it aligns neatly with the steps of the READ approach: the method usually involves familiarisation (eg, by reading), coding (ie, labelling aspects of the text), developing a framework (ie, a set of labels indicating key information), applying that framework to all documents, charting data (ie, summarising the data from each of the documents into a spreadsheet) and finally, interpreting the data. We applied this last stage of the Framework Method to analyse our dataset as a whole. As part of our analysis, we identified, through discussion, the implications of our findings, as well as recommendations for where improvements are needed.

### Distill your findings

This final step involved refining the findings and presenting them, organised by category, illustrated by key extracts and examples.

### Patient and public involvement

Patients and the public were not involved in the document analysis. However, three survivors were part of the RASDIH expert advisory group and shaped the wider project's findings.

## FINDINGS

We found 13 guidance documents on DVA and/or medical confidentiality and information sharing more generally, summarised in table 1. Publication dates were between 2003 and 2020. Eleven documents were primarily for HCPs, while two[37 38] mainly targeted staff working in safeguarding (often social care professionals). One document, from the GMC,[39] specifically targeted doctors; another from the NMC[40] targeted nurses, midwives and nursing associates. In table 1, document number 5 was written by a Caldicott council member with DVA expertise and documents 4 and 8–11 were written with input from clinical academics with DVA expertise. Table 2 summarises the three laws and duties that the guidance documents cited and that legislate on when sharing personal information is allowed without consent. The Mental Capacity Act 2005[41] is also relevant in that information can be shared without consent if a person lacks the capacity to make a decision related to that information, but guidance documents did not specifically cite this act in relation to information sharing.

The documents provided overarching guidance on (1) sharing information without consent; and specific guidance on (2) sharing information with or for MARACs; (3) sharing information for formal safeguarding; and (4) sharing information within the NHS. We present our findings under these headings below.

### Sharing information without consent: overarching guidance

The 2014 BMA report on DVA[42] states that 'a refusal to disclose information by a competent adult can be overridden in order to protect a third party, such as a child or vulnerable adult, who may be in the household'

**Table 1** Guidance on DVA and/or confidentiality and information sharing from national bodies in England

| National body | Guidance title | Stated target audience | Stated purpose |
|---|---|---|---|
| Adass and Local Government Association[37] | Adult safeguarding and domestic abuse: a guide to support practitioners and managers | For practitioners and managers in local authorities (ie, social care, which has responsibility for safeguarding) and partner agencies (which can include the healthcare service) | To help staff to give better informed and more effective support to people who need an adult safeguarding service because of DVA |
| BMA[45] | Adults at risk, confidentiality, and disclosure of information guidance | All doctors | To set out the current legal and ethical position on disclosure of information relating to adults who retain capacity but may be subject to some form of duress |
| BMA[46] | Adult safeguarding toolkit | Principally for doctors, but also useful for any professional working in health | To explain doctors' role in safeguarding adults who may be at risk of abuse or neglect |
| BMA[42] | Domestic abuse. A report from the BMA Board of Science | Not stated (but BMA guidance is generally for doctors) | Not stated |
| UK Caldicott Guardian Council and Department of Health[52] | 'Striking the Balance' Practical guidance on the application of Caldicott guardian principles to domestic violence and MARACs | Attendees of MARACs (which can include HCPs) and Caldicott Guardians* | Best practice guideline to assist the decision-making of those involved in sharing information about DVA between agencies |
| DH[47] | Confidentiality NHS code of practice | NHS staff, Caldicott guardians, data protection officers and anyone working in and around health | Best practice guideline on required practice for those who work within or under contract to NHS organisations concerning confidentiality and patients' consent to the use of their health records |
| DH[48] | Confidentiality: NHS Code of Practice Supplementary Guidance: Public Interest Disclosures | Medical directors, directors of nursing, local authorities, directors of adult social services, directors of human resources, allied health professionals, GPs, communications leads, emergency care leads and directors of children's social services | Expands on the principles set out in the 2003 DH guidance above and aims to aid staff in making difficult decisions about when disclosures of confidential information may be justified in the public interest |
| DH[51] | Responding to domestic abuse: a resource for health professionals | Primarily all NHS staff. Also helpful for staff in partnership agencies who work with adults and children and healthcare commissioners and managers. | To support continuous improvement in the health service response to DVA |
| General Medical Council[39] | Confidentiality: good practice in handling patient information | Doctors | Sets out the principles of confidentiality and respect for patients' privacy that doctors are expected to understand and follow |
| National Institute for Health and Care Excellence (NICE)[30] | Domestic violence and abuse: multiagency working. Public health guideline | Health and social care professionals, commissioners and providers, specialist DVA services, criminal justice and detention centre staff, people affected by DVA and their families and carers and members of the public | To help identify, prevent and reduce DVA |
| NICE[67] | Domestic violence and abuse: quality standard | As above | Specific, concise, and measurable statements to be read in conjunction with NICE guidelines |

Continued

**Table 1** Continued

| National body | Guidance title | Stated target audience | Stated purpose |
|---|---|---|---|
| Nursing and Midwifery Council[40] | The code: professional standards of practice and behaviour for nurses, midwives, and nursing associates | Nurses, midwives and nursing associates | Professional standards that these registered HCPs must uphold |
| Social Care Institute for Excellence (SCIE) with DH, Adass and Local Government Association[38] | Safeguarding adults: sharing information | Front-line staff and volunteers—unclear from which sectors—likely social care since SCIE is a social care body. | Support implementation of the adult safeguarding aspects of The Care Act 2014 |

*A Caldicott guardian is a senior person within a health or social care organisation who makes sure that personal information is used legally, ethically and appropriately, and that confidentiality is maintained. All NHS organisations and local authorities that provide social services must have a Caldicott guardian.
BMA, British Medical Association; DVA, domestic violence and abuse; GPs, General Practitioners; HCPs, healthcare professionals; MARACs, multiagency risk assessment conferences; NHS, National Health Service.

(p.54). It also states that decisions about sharing information become more difficult when an adult refuses to disclose information to protect themselves rather than a third party. Here, the HCP faces an ethical tension: they can keep confidence to respect the patient's autonomy, potentially increasing the risk of harm to the patient; or they can prioritise their perceived duty of care and beneficence but act in a way that curtails the patient's autonomy. The guidance does not elaborate on this tension nor on how to resolve it.

The 2014 BMA[42] recommendation is similar to the GMC's recommendation from its 2017 guidance on medical confidentiality (not DVA specific),[39] although while the BMA[42] does not cite any specific law, the GMC cites the Crime and Disorder Act 1998[43] and General Data Protection Regulation.[44] Paragraph 9 in the GMC guidance states that the law permits sharing information without consent in some situations, including if the patient lacks capacity and sharing would be to their overall benefit (beneficence) or could be justified in the public interest. The GMC defines public interest as a risk to others and in fact uses DVA as an example of when HCPs can share information in the public interest ('When victims of violence refuse police assistance, disclosure may still be justified if others remain at risk, for example … domestic violence when children or others may be at risk' (paragraph 65, p.34).) Thus, like the BMA,[42] the GMC makes clear that a patient's information can be shared to protect a third party.

The GMC[39] is somewhat clearer than the BMA[42] on what to do when no third party is at risk: '[Y]ou should usually abide by the patient's refusal to consent to disclosure, even if their decision leaves them (but no-one else) at risk of death or serious harm' (paragraph 37, p.32). In such cases, the GMC's default position is to respect and prioritise the patient's autonomy over their beneficence, and to not share information. However, the qualifier 'usually' in this recommendation is significant. The recommendation also has a reference to an endnote, which states that HCPs can share a patient's information without their consent even if they have capacity, but in restricted circumstances. Later BMA guidance from 2018 and 2020[45 46] similarly states, and sets out when, HCPs can share information without consent for a patient who has capacity. In so doing, it fills a gap in their 2014 report. In table 3, we compare the later BMA and GMC wording (from the endnote) on when sharing information without a patient's consent is permissible.

Both bodies recommend that information can be shared when no third party is at risk, that is, when the harm in question is to the individual themselves (although the BMA[45 46] does this less explicitly). Both state that cases should be exceptional and that there should be evidence of imminent risk. However, on a close reading, subtle differences in these recommendations are apparent. The BMA wording seems to require that there be evidence of risk of crime; the GMC's[39] wording suggests the evidence should pertain to the risk of harm. The BMA wording

**Table 2** Laws on confidentiality

| Crime & Disorder Act 1998 section 115[43] | Common law duty of confidentiality | General Data Protection Regulation/Data Protection Act 2018[44] |
|---|---|---|
| Any person may disclose information to a relevant authority 'where disclosure is necessary or expedient for the purposes of the Act (reduction and prevention of crime and disorder)'. Relevant authorities, broadly, are the police, local authorities (eg, safeguarding), health authorities and local probation boards. | Common law generally allows the disclosure of confidential information if<br>► the patient consents<br>► it is required by law, or in response to a court order<br>► it is justified in the public interest. | Relevant lawful bases for processing information—set out in article—are<br>► legal obligation: you can rely on this lawful basis if you need to process the personal data to comply with a common law or statutory obligationvital interests: you are likely to be able to rely on vital interests as your lawful basis if you need to process the personal data to protect someone's life. |

states that disclosure should be likely to prevent the harm (or the crime—this is ambiguous), whereas the GMC's states nothing about likelihood, but rather that no other methods of preventing the harm should exist. Both bodies guide doctors, but they give slightly different recommendations; thus, it remains unclear in what circumstances sharing information is permissible.

### Ambiguous terms

The meaning of 'serious crime' and 'serious harm' (eg in the statements in table 3) is also unclear and therefore so is the threshold at which autonomy becomes less important than other ethical principles. The GMC[39] gives some examples of serious crimes: murder, manslaughter or serious assault, while the earlier 2003 NHS code of practice on confidentiality[47] recognises that 'the definition of serious crime is not entirely clear' and also mentions 'rape…kidnapping, child abuse or other cases where individuals have suffered serious harm' as examples. This

2003 code of practice[47] states that these types of crime 'may all warrant breaching confidentiality' (p.34). In their 2010 supplementary guidance to this code of practice,[48] the DH adds that serious crimes include those 'that cause serious physical or psychological harm to individuals … and will likely include other crimes which carry a five year minimum prison sentence but may also include other acts that have a high impact on the victim' (p.9). DVA will often have a high impact on the victim-survivor. Notably, despite coercive control becoming an offence in 2015 as part of the Serious Crime Act,[49] it is not mentioned in definitions of serious crime, nor anywhere else in these (including post-2015) documents. Whether HCPs recognise the high impact of DVA and that relevant laws are in place will depend on their understanding of DVA.

Neither the 2014 NICE guidelines[30] nor the 2018 NMC code for professional standards[40] mention serious harm or serious crime. Rather, they use the terms 'serious risk' and 'risk of harm' without qualifiers: both bodies thus give broader guidance than the 2017 GMC[39] and 2018 and 2020 BMA documents.[45 46] Specifically, NICE[30] states that 'information should be shared only with the person's consent unless they are at serious risk' (p.14), while the NMC simply recommends that nurses 'share information if you believe someone may be at risk of harm, in line with the laws relating to the disclosure of information' (17.2, p.15). No laws are specified.

As hinted at above, across all guidance documents, there is ambiguity in what 'the public interest' means: specifically whether the public interest test applies when only the victim-survivor is at risk. The 2019 SCIE safeguarding adults guide,[38] which outlines appropriate sharing between local authorities and the health service, interprets 'the public interest' as a risk to third parties: ('Make sure that *others* are not put at risk by information being kept confidential: Does the public interest served by disclosure of personal information outweigh the public interest served by protecting confidentiality?' (p.19, emphasis added)). In her analysis of GMC guidelines, Cave points out that since 1977, the organisation has moved between different definitions of the public interest: in some guideline editions, the public interest has meant 'only third parties'. In other editions, it has meant 'third parties or only the patient'.[50] As we have

**Table 3** When is sharing information without a competent patient's consent, or in the face of withheld consent, permissible?

| BMA adults at risk, confidentiality, and disclosure of information guidance[45] and BMA adult safeguarding[46] | GMC confidentiality guidance[39] |
|---|---|
| 'Disclosure of information without consent … is likely to be exceptional. This is likely to be where there is strong evidence of a clear and imminent risk of a serious crime likely to result in serious harm to the individual, and the disclosure of the information is likely to prevent it'. | 'In very exceptional circumstances, disclosure without consent may be justified in the public interest to prevent a serious crime such as murder, manslaughter or serious assault even where no-one other than the patient is at risk. This is only likely to be justifiable where there is clear evidence of an imminent risk of serious harm to the individual, and where there are no alternative (and less intrusive) methods of preventing that harm' (endnote 18, p.73). |

BMA, British Medical Association.

discussed, the most recent guidance from 2017[39] states that the public interest is usually relevant only when third parties are at risk but in an endnote, it states the public interest may also be relevant when only the patient is at risk, although in very exceptional circumstances. The GMC[39] states that this is an uncertain area of law, and that HCPs should seek legal advice before making disclosures on these grounds. The 2017 DH resource[51] also states that disclosures can be made without consent in the public interest and uses a definition (stated earlier) that encompasses cases where only the patient is at risk. But it introduces ambiguity because it cites the 2003 NHS code of practice on confidentiality,[47] the 2010 supplementary guidance of which[48] provides examples of public interest defences that include harm to third parties only. Cave's[50] analysis of this 2010 guidance is that 'patients' best interests will not justify disclosure if they have capacity, and neither will the public interest, except to 'prevent serious harm or death *to others*". (p.18, emphasis added). That is, public interest applies only when third parties are at risk: so, when the patient has capacity, information should generally not be shared. The 2017 DH resource[51] on DVA and its earlier guidance on confidentiality[48] are thus inconsistent. Overall, it takes a close reading of all recent documents for it to become apparent that the public interest test can encompass exclusive risk to the victim-survivor.

### Inconsistencies

The 2017 DH resource[51] makes two inconsistent recommendations within the same document. It states that HCPs can share information in responding to victim-survivors if sharing 'can be justified in the public interest, such as where there is a risk of harm to the victim, any children involved or somebody else if information is not passed on' (p.43). On documenting information in perpetrators' records, it states 'while these records are strictly confidential, if there is a risk of death to an adult or a risk of significant harm to a child, this will override any requirement to keep information confidential' (p.54). The documents indicate that perpetrators' information can be shared when there is a risk of death to an adult or risk of significant harm to a child, while victim-survivors' information can be shared when there is a risk of harm (without the qualifier 'significant') to the victim-survivor, child or somebody else. Thus, the recommendations give the impression that sharing confidential information about perpetrators can happen in more restricted situations than sharing information about victim-survivors.

### Specific guidance on sharing information with or for MARACs

GMC guidance[39] and the DH resource,[51] both from 2017, provide guidance for HCPs regarding referrals to MARACs. The GMC states, 'personal information may be disclosed to a MARAC with consent, or if the disclosure can be justified in the public interest' (paragraph 21, p.73). Given that the GMC defines the public interest as applying when third parties are at risk (except in 'very

exceptional' circumstances), it is plausible that a HCP may understand from this that they should generally disclose a victim-survivor's information to a MARAC only when the victim-survivor consents to this sharing. The DH resource[51] in fact states, 'You will need the consent of a competent adult victim to refer them to a MARAC, unless the public interest test is engaged with the high threshold risk' (p.36).

Interestingly, the 2019 SCIE guide[38]—which differs from other documents since it is for social care professionals primarily—states that local authority professionals can share information with MARACs without consent. DH and Caldicott guidance[52] on sharing information for MARACs from 2012 also makes this point: it states that although victim-survivors should usually be told about the referral to a MARAC, 'consent is not asked for, because the decision has already been taken that a MARAC is needed, based on the risk to the victim' (p.6). This guidance emphasises that since it is high-risk cases that are referred to MARACs, information sharing without consent is justified. It highlights that a professional responsibility to share information can in some circumstances outweigh the duty of confidentiality owed to the individual.

### Specific guidance on sharing information for formal safeguarding

Cases where victim-survivors are vulnerable (as per The Care Act 2014[18]) and/or have children under 18 will usually fall under formal safeguarding protocols. The SCIE[38] guide outlines when a professional can override a person's refusal to consent to information sharing with safeguarding partners when formal safeguarding applies: when the "alleged abuser" also has care and support needs or when the person has capacity but may be under duress or being coerced. Notably, this is the only guidance to explicitly state that coercion can affect a person's decision-making, even if they 'have capacity.'

### Sharing information within the health service

The 2014 BMA[42] report emphasises that DVA is a multidisciplinary concern and needs a joined-up approach across teams. However, no guidance states whether and how HCPs should share DVA information within the health service. NICE and 2017 DH guidance does, however, suggest that HCPs should offer referrals, or example, to substance use treatment, mental health services and sexual assault referral centres.[30 51] The DH moreover clarifies that if the HCP to whom the patient has disclosed is not a GP, the HCP should refer the patient to their GP, who can refer them for onwards mental health support.[51]

### DISCUSSION

This study is the first to explore and analyse guidance documents for HCPs (and social care professionals) around information sharing relating to DVA. Documents primarily focused on sharing information without consent and on specific examples of such sharing, that is,

for MARACs and safeguarding. A striking initial finding was that 13 different documents all aim to guide good practice. Although one targeted doctors (GMC[39]) and one targeted nurses, midwives and nursing associates (NMC[40]), professionals mostly work in multidisciplinary teams and make decisions together: teams thus draw on a large number of partly conflicting guidelines.

A key finding is that the documents emphasised sharing information without consent should be exceptional. The DH[51] and GMC[39] even recommend that MARAC referrals should happen without consent only in exceptional situations. Overall, the guidance documents emphasise consent and confidentiality as default positions, a finding that points to a tendency among HCPs to prioritise autonomy when weighing it against beneficence.[53] Respecting autonomy is, of course, *crucial*: in cases of DVA it means not reproducing the abusive dynamic between victim-survivor and perpetrator, and not further undermining the victim-survivor's agency. At the same time, the guidance documents are problematic because they conflate 'respecting autonomy' with 'keeping information confidential'. In other words, the documents embed assumptions that victim-survivors do not want their information shared. Olive[26] found that victim-survivors who had sought emergency care due to a partner assault thought emergency department HCPs would share information about DVA with their GPs as routine practice. However, in 80% of the medical records analysed, clinicians did not mention DVA or assault at all in discharge letters to GPs. In some cases, victim-survivors will indeed request their information not be shared. They may be frightened of the consequences, feel unprepared for others to intervene, or think sharing is unsafe: victim-survivors are sometimes the best judges of their safety. However, in some cases, when the risk is high, it will be appropriate for HCPs to share information even when a victim-survivor has refused consent or when the HCP has not managed to seek consent. Sharing in such cases can sometimes be justified by beneficence. For example and as indicated earlier, in practice, MARAC does not require victim-survivors' knowledge, let alone consent, because they are instrumental in protecting them from harm.[52]

A second key finding is that none of the documents, except SCIE's 2019 safeguarding guide,[38] mention that coercive control can hinder patients' autonomous decision-making. Stark,[5] whose work has been fundamental to the understanding of coercive control, argues that because it attacks a victim-survivor's autonomy and liberty, it 'compromises the(ir) capacity for independent, self-interested decision-making' [p.3]. Stark is critical of professionals who assume victim-survivors can exercise decisional autonomy 'between episodes' of DVA. Specifically, he emphasises that coercive control does not happen in discrete episodes but is an ongoing form of entrapment, and that autonomous decision-making is made unlikely because perpetrators deprive victim-survivors of the material and cognitive resources needed for such decision-making. A patient experiencing coercive

control might have mental capacity but whether they have 'autonomy' is questionable: they may say they do not want their information shared because the level of abuse has induced compliance to the perpetrator. Coercive control, therefore, complicates the tension between autonomy and beneficence. But guidance documents from most national bodies in England stop at stating that if a patient has capacity to make a decision as defined by The Mental Capacity Act 2005,[41] HCPs should generally respect their decision, even if they think it unwise. (According to the Act, unless a person's impaired decision-making is caused by an 'an impairment of, or disturbance in, the functioning of the mind or brain', eg, a psychiatric illness, learning disability, dementia, brain damage, the Act does not engage.) These findings may reflect the understanding of policy-makers and HCPs who draft these documents—specifically, a limited understanding of the manifestations, dynamics and effects of coercive control. Cohen and Caswell[54] criticise the 2017 GMC[39] guidance for not supporting confidentiality breaches in cases where victim-survivors (who are adult, and have no children under 18, so do not fall under safeguarding duties) are facing high-risk abuse but decline information sharing or referral. Domestic homicide reviews[8 17] and safeguarding adults reviews[16] have indeed made clear that HCPs, police and social care have made poor decisions not to share information with others about DVA because the patient had the capacity to withhold consent and because no formal safeguarding duties applied. In these cases, the decision to share information might have prevented a homicide.

These two key findings highlight that this is a complex area and requires a position that is more nuanced than an automatic prioritisation of confidentiality and a default reliance on explicit consent. The elevation of autonomy and explicit consent is a welcome important move away from paternalism, that is, acting in an autonomous person's interest without taking their will into account. But in a bid to move away from paternalism, beneficence may be lost. A novel lens through which to think about these tensions is 'maternalism': Sullivan[55] defines maternalism as acting in a way that is thought to be in line with an autonomous patient's will and motivated by a desire to improve the patient's welfare, although not based on the patient's expression of consent or assent. She argues that some actions that are assumed to be paternalistic are actually maternalistic. While paternalism describes a general kind of medical judgement, maternalism is based on an interpersonal relationship of trust and understanding between a HCP and patient. This relationship can help the HCP understand and make decisions in line with the patient's interests, desires and values. Importantly, though, when sharing information is done badly or when victim-survivors' understandings of what is safest for them is ignored, it may cause victim-survivors to lose trust, disengage from services and face a higher risk.

A third key finding from our analysis is that when the guidance documents explain the exceptional cases in

which sharing can be done without consent—that is, when beneficence can trump autonomy—explanations between, and sometimes within, documents are inconsistent, contradictory and ambiguous. HCPs and other professionals may be unsure which guidance document to follow. Ambiguous terms and concepts may also have this effect. 'The public interest' is an unclear concept, perhaps because no agreed on mechanism exists for establishing what is 'in the public interest' despite calls for consistency and clarity around the term in DVA cases.[56] 'Harm' is also ambiguous and requires an understanding of the nature and health consequences of DVA (particularly non-physical abuse), which some professionals, including HCPs, do not have.[5 57 58] At the same time, this ambiguity leaves room for professional judgement, which is appropriate. Some of the ambiguity including around the public interest may be due to intrinsic ethical conflicts, but guidance documents do not make these conflicts explicit.

Guidance is especially lacking on whether and how HCPs should share DVA information within the health service, for example, between GPs and other HCPs who are already providing care to the patient. Guidance on what to share about perpetrators is completely lacking, except in one document that states sharing perpetrator information should be especially exceptional.[51] Both are areas for improvement: analyses of domestic homicide reviews[6–9 11 12] make clear that information, particularly information about perpetrators, is inadequately shared between different parts of the health service.

Looking across these documents, the need for consistency and coherence in the healthcare response to DVA is obvious. Too many different guidance documents exist, and HCPs arguably need one set of recommendations to implement good practice. However, it is unclear how much inconsistencies between guidance documents affect clinical practice. Variation between contemporaneous clinical guidelines has been recognised for decades[59] and there is evidence from primary care that guidelines are rarely consulted by clinicians.[60] No research explores how HCPs share information about DVA with other HCPs, and few studies[23 25 61] explore sharing between health and other agencies or services. As such, it is unclear how much HCPs rely on guidance documents in practice, whether they rely more on the local norms within their teams and specialities and how much these norms and guidelines align. Research from other areas of healthcare on information sharing suggests that HCPs are not always sure what guidelines exist, what guidelines state or how to implement them: they resultantly make conservative assumptions about what sharing is permitted. The research indeed also suggests that they rely more on norms and professional judgement than published guidance.[62–66] This is a limitation of our work, in that we are unsure of whether the issues we have identified have any significant impact on practice. More specific limitations include that our method was not a systematic review, and we were just two analysts, which may have limited the rigour of our analysis.

## Conclusion and recommendations

In England, national guidance for HCPs on sharing information about DVA is numerous, inconsistent, ambiguous and lacking in detail. There is a need for coherent guidance for cross-speciality clinical practice: in our wider project, an expert advisory group developed such a set of guidance for all HCPs.[33]

In terms of future practice and policy, we recommend that HCPs ought to take coercive control and its relationship with capacity into account when deciding what is in the interest of the victim-survivor.[54] We also recommend that development groups for guidelines and recommendations on the healthcare response to DVA should include professionals with expertise in the manifestations, dynamics and effects of coercive control and other forms of DVA, as well as experts by experience. Finally, we recommend that more work is done to improve practice around recording and sharing information about DVA within healthcare, beyond the production of good practice guidance, including research to evaluate what impact, if any, such guidance has on recording and sharing DVA information. Perhaps more importantly, since HCPs may rely more on local norms and professional judgement over guidance documents for decision-making, we need fora for HCPs to discuss dilemmas and difficulties they encounter in responding to DVA.

**Correction notice** This article has been corrected since first published online. Pre-production team error has been rectified and the term "victim-survivor" has been made consistent throughout. The open access licence has been updated to CC BY.

**Acknowledgements** Thank you to members of the Pathfinder consortium; thank you to Dr Kathryn Pitt for her work in identifying research questions; and thank you to all members of the expert advisory group particularly the experts by experience and Christopher Fincken for comments on an early version.

**Contributors** GF conceived of the broader aims of the Recording and Sharing Information about DVA in Healthcare project. SD developed the project's methodology and led the design, analysis and data interpretation of the presented work. SD wrote the manuscript with critical input from GF. Both authors approved the final manuscript. SD is responsible for the overall content as the guarantor.

**Funding** The RASDIH work was supported by funding from the Pathfinder consortium (Standing Together Against Domestic Abuse, Safelives, AVA, IRISi, Imkaan) which was in turn funded by Department of Health and Social Care and Department of Digital, Culture, Media and Sport, UK. The dissemination, including this article, was supported by a grant from the University of Bristol UKRI Quality-Related Strategic Priorities Fund [no grant numbers].

**Competing interests** GF chaired NICE (2014) Domestic Violence and Abuse guideline development group and made a contribution on working with children in families affected by DVA in the DH resource (2017).

**Patient and public involvement** Patients and/or the public were involved in the conduct of this research. Refer to the Methods section for further details.

**Patient consent for publication** Not applicable.

**Provenance and peer review** Not commissioned; externally peer reviewed.

**Data availability statement** No data are available. Data sharing not applicable as no datasets were generated and/or analysed for this study. All guidance documents analysed are available online.

**ORCID iDs**
Sandi Dheensa http://orcid.org/0000-0002-6412-696X
Gene Feder http://orcid.org/0000-0002-7890-3926

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
