## [Reviewer comments · BMJ Open]

ARTICLE DETAILS

TITLE (PROVISIONAL)	Sharing information about domestic violence and abuse in healthcare: an analysis of English guidance and recommendations for good practice.
AUTHORS	Dheensa , Sandi; Feder, Gene

VERSION 1 – REVIEW

REVIEWER	Patricia Easteal University of Canberra
REVIEW RETURNED	26-Dec-2021

GENERAL COMMENTS	Excellent paper. I have reviewed many manuscripts over the course of a lengthy academic career and normally have many suggestions for revision. However, you covered all potential concerns that I had except one. Those you addressed that I was anticipating would need to be looked at were: 1)problems/issues in defining 'imminent risk'; the absence of coercive control in documents; 3) that a 'close reading of all documents for it to become apparent that public interest can encompass exclusive risk to the victim/survivor; 4). your finding that 'none of the documents, except SCIE's 2019 safeguarding guide [29], mention that coercive control can hinder patients' autonomous decision-making.'. These findings are raised in the discussion and conclusion, which is excellent. The only area/thought missing is a bit peripheral to this research but I think it's important to appear in the Conclusion. To me, the research findings are evidence of how vitally important it is that health providers (and those in Government drafting policy and legislation) who are often the first port of call for FV victims to possess an in-depth understanding of the manifestations, dynamics and effects of FV. They need such knowledge to enable them to 'see' imminence' the issues of 'consent' and coercive control
---

REVIEWER	Roxanne Keynejad King's College London, Institute of Psychiatry
REVIEW RETURNED	08-Feb-2022

GENERAL COMMENTS	Thank you for the opportunity to review this interesting paper on an important subject. In its current form, I believe this paper to require major revisions for the reasons I outline below. Primarily, the limitations arise from a lack of clarity about the methods (is it a qualitative study? What is the READ method? Is it a rapid review? A desk review?) and a disconnection between the findings and the discussion, with recommendations that appear to follow much more from other, related work done for a related but distinct study. I would be happy to review a revision of this work as it is an important contribution to our knowledge of this field.
---

	The paper would benefit from more consistent referencing of studies when they are mentioned and more clarity about the UK-specific focus of the work for an international audience. ABSTRACT Please define 'READ' How were the documents identified? What is the wider project in which this article sits? It is a bit confusing for the reader that recommendations from that work (presumably, published elsewhere?) are presented. STRENGTHS AND LIMITATIONS I wonder whether a term like 'communities of practice' will be understood outside medical education circles, without definition? INTRODUCTION Personal preference: I find it preferable to say "or" or "and" rather than "/" in formal prose. As in, "domestic violence and abuse" rather than having a "/" between them. Ditto for "victims/survivors". The definition of DVA requires a reference. Perhaps state which countries the NHS pertains to, for international readers? The statement "no-one responds" is a generalisation and could be phrased in more conditional language. Top of p4. Which analyses? Can this be referenced? Please also reference the statements about DHRs and intimate partner femicide. p4-5 re: care and support needs/safeguarding: perhaps clarify that these statements refer specifically to procedures in specific countries, for international readers? Please reference the statement about best practice around information sharing. Table 1: It isn't typical to present tables in the Introduction. Perhaps this could be a supplementary file? METHOD As with the abstract, please define and explain the READ approach. Can the authors offer a description of their search methodology? Was this a rapid review? A desk review? Might the authors provide a flow diagram showing how many potentially eligible results were screened? Can the authors list all the professional bodies whose websites were searched? What was done in cases where the websites did not yield results (e.g. required password access or had poor search engine functionality)? Were HCPs consulted? Please provide references for the methods used, such as the framework approach. The sentence "questions such as whether... facilitated data analysis" is vague. In what way did they facilitate data analysis? Were they rows in the framework matrix? Some other way? The methods read like a qualitative study but if so, they would benefit from a statement on positionality (were analysers clinicians themselves?), reflexivity and other aspects of qualitative research practice. Can the wider project be named? Can a reference to the study website be provided or a protocol paper or a study registry link? FINDINGS This section reads as though the authors are presenting themes from qualitative synthesis of the texts analysed but this terminology
--	--

	is not applied. It might harmonise the paper if consistent terminology (either uniform discussion of methods and themes or another set of terms more suited to the READ method?) were used across the methods and findings. Table 3: please define GDPR when it is first used. The early paragraphs could be more concise by removing the lengthy direct quotations, which are in any case repeated in Table 4. p15 please specify and reference which code of practice is referred to here. p17 The sentence beginning "perpetrators' information..." is unclear. Are the authors stating this as a fact or paraphrasing what this document says? p19 The sentence beginning "the DH moreover clarifies" needs a reference, given there are multiple DH references. DISCUSSION p19 Perhaps it is not true to say HCPs are faced with 13 documents, given that midwives are unlikely to read BMA guidance and vice versa? p20 and p21 the direct quotes in the Discussion could be phrased in the authors' own words. p24 I don't clearly follow the meaning of this final paragraph beginning "the aim of the research presented here..." CONCLUSIONS Please state the countries involved for international readers, rather than saying "national". The summary of other work not presented in this paper does not belong under the conclusions of this paper. The contents of Table 5 do not follow from the results presented. Perhaps they are more helpful as a supplementary resource? The conclusion that sharing without consent be done in a maternalistic way does not follow from the study presented. Perhaps it belongs in the other study?
--	---

VERSION 1 – AUTHOR RESPONSE

Reviewer 1: Dr. Patricia Eastal, University of Canberra

Excellent paper. I have reviewed many manuscripts over the course of a lengthy academic career and normally have many suggestions for revision. However, you covered all potential concerns that I had except one. Those you addressed that I was anticipating would need to be looked at were:

1) problems/issues in defining 'imminent risk'; the absence of coercive control in documents; 3) that a 'close reading of all documents for it to become apparent that public interest can encompass exclusive risk to the victim/survivor; 4). your finding that 'none of the documents, except SCIE's 2019 safeguarding guide [29], mention that coercive control can hinder patients' autonomous decision-making.'. These findings are raised in the discussion and conclusion, which is excellent.

Thank you so much for this positive feedback!

The only area/thought missing is a bit peripheral to this research but I think it's important to appear in the Conclusion. To me, the research findings are evidence of how vitally important it is that health providers (and those in Government drafting policy and legislation) who are often the first port of call for FV victims to possess an in-depth understanding of the manifestations, dynamics and effects of FV. They need such knowledge to enable them to 'see' imminence' the issues of 'consent' and coercive control

This is an excellent point thank you. Since our paper was about policy, not practice, we have integrated your helpful insight as follows:

- We have added to our Abstract *Coherent recommendations for cross-speciality clinical practice are needed. Recommendations should reflect an understanding of the manifestations, dynamics, and effects of DVA, particularly coercive control.*
- We have added to our Findings: *In table 2, reference number 4 was written by a Caldicott council member with DVA expertise and 5 and 8-11 were written with input from clinical academics with DVA expertise*
- We have added to our Discussion *Variation between contemporaneous clinical guidelines has been recognised for decades [60] and there is evidence from primary care that guidelines are rarely consulted by clinicians ... These findings may reflect the understanding of policymakers and HCPs who draft these documents—specifically, a limited understanding of the manifestations, dynamics, and effects of coercive control.*
- We have added to our Conclusion *We also recommend that development groups for guidelines and recommendations on the healthcare response to DVA should include professionals with expertise in the manifestations, dynamics, and effects of coercive control and other forms of DVA, as well as experts by experience.*

Reviewer 2: Dr. Roxanne Keynejad, King's College London

Thank you for the opportunity to review this interesting paper on an important subject. We are glad that you think the paper is interesting and important – thank you

In its current form, I believe this paper to require major revisions for the reasons I outline below. Primarily, the limitations arise from a lack of clarity about the methods (is it a qualitative study? What is the READ method? Is it a rapid review? A desk review?)

We hope we have now addressed these concerns.

We now explain below that the READ approach (Dalglish et al.) is an approach to document analysis. This approach is a method of qualitative health policy research and involves four-steps for gathering, and extracting information from, documents. Its four steps are: (1) Ready your materials, (2) Extract data, (3) Analyse data and (4) Distil your findings. It is a desk review, which we now state, but it is not explicitly called a 'desk review' in Dalglish et al.'s paper.

and a disconnection between the findings and the discussion, with recommendations that appear to follow much more from other, related work done for a related but distinct study.

We hope we have now addressed these concerns.

As an explanation, this document analysis was one part of a bigger study (Recording and Sharing Information about DVA in Healthcare – R&SDIH – the report is in the references), which itself was part of the Health Pathfinder project <https://www.standingtogether.org.uk/pathfinder> . which aimed to enhance the healthcare response to DVA. We briefly explain this in the Aim.

As we state below, we have deleted the recommendations and summary from the bigger study.

The paper would benefit from more consistent referencing of studies when they are mentioned. We have now ensured our references are consistent.

And more clarity about the UK-specific focus of the work for an international audience. We have now clarified that we mean England when we use the word 'national.'

I would be happy to review a revision of this work as it is an important contribution to our knowledge of this field.

Thank you, we are glad you think so and appreciate your suggestions for improvement. .

ABSTRACT

Please define 'READ'

How were the documents identified?

We have stated the following:

We conducted a desk-based review, adopting the READ approach to document analysis. This approach is a method of qualitative health policy research and involves four-steps for gathering, and extracting information from, documents. Its four steps are: (1) Ready your materials, (2) Extract data, (3) Analyse data, and (4) Distil your findings. Documents were identified by searching websites of national bodies in England that guide and regulate practice and by backwards citation searching documents identified initially.

What is the wider project in which this article sits? It is a bit confusing for the reader that recommendations from that work (presumably, published elsewhere?) are presented.

We have now deleted the part of the abstract where we mention this wider project, but we explain it later.

STRENGTHS AND LIMITATIONS

I wonder whether a term like 'communities of practice' will be understood outside medical education circles, without definition?

We have changed our strengths and limitations section in line with editor comments so have deleted this term and replaced it with 'teams and specialities' when we use it later.

INTRODUCTION

Personal preference: I find it preferable to say "or" or "and" rather than "/" in formal prose. As in, "domestic violence and abuse" rather than having a "/" between them. Ditto for "victims/survivors". We have changed the /'s to 'or' except in 'and/or,' which is accepted parlance. We have changed victim/survivors to victim-survivors in line with the language used in the Women's Aid Research Integrity Framework <https://www.womensaid.org.uk/evidence-hub/research-and-publications/research-integrity-framework/>

Definition of DVA requires a reference.

We have added the UK government definition as a reference.

Perhaps state which countries the NHS pertains to, for international readers?

We have added 'UK's National Health Service' and used the term 'national.'

The statement "no-one responds" is a generalisation and could be phrased in more conditional language

We have changed this to "no-one responded" (and changed the 'has' earlier in the sentence to 'had' to make clear that we are describing DHR findings).

Top of p4. Which analyses? Can this be referenced?

We meant the same analyses that we mentioned in the paragraph before. We have now reordered this section to make this point clearer.

Please also reference the statements about DHRs and intimate partner femicide.

We have now added references throughout. Please note that the references for the part beginning "The information in question..." are in the first line of the paragraph. We looked across all these reviews of DHRs and these were the common themes.

p4-5 re: care and support needs/safeguarding: perhaps clarify that these statements refer specifically to procedures in specific countries, for international readers?

We have stated “*In the UK, formal safeguarding processes apply...*” and we have referenced the Care Act 2014.

Please reference the statement about best practice around information sharing.

We have now edited this statement to make it more precise and have added a reference to GDPR. It now reads: “*Sharing information with consent is a lawful basis for sharing information.*”

Table 1: It isn't typical to present tables in the Introduction. Perhaps this could be a supplementary file?

We have deleted the table and edited the text preceding it so that the key points are now in text.

For example, referral to safeguarding or MARAC does not require consent or even victim-survivors' knowledge [20,21]. More generally, sharing health-related information within clinical teams, which can include a wide range of staff who have not met the patient, can happen with implied consent. Sharing information with NHS-based but externally employed DVA workers or agencies and third-sector mental health or substance services would usually rely on explicit consent [22].

METHOD

As with the abstract, please define and explain the READ approach.

The READ approach has been developed Dalglish et al, which we reference. We have added:

We conducted a desk-based analysis of guidance documents, adopting the READ approach [35] to analyse the documents. This approach is a method of qualitative health policy research and involves four-steps for gathering, and extracting information from, documents. Its four steps are: (1) Ready your materials, (2) Extract data, (3) Analyse data, and (4) Distil your findings.

Can the authors offer a description of their search methodology?

We have now added the following:

Ready your materials

Within the READ approach, researchers first establish parameters around the topic, where to search, and dates. We searched for documents that guided practice around medical confidentiality and information sharing, or DVA specifically. We decided to focus on the policies of national bodies from England that guide and regulate healthcare across specialities. With R&SDIH's professional advisory group (comprising clinicians, medical confidentiality experts, 'DVA and health' researchers, and DVA practitioners), we decided upon the British Medical Association (BMA), General Medical Council (GMC), Nursing and Midwifery Council (NMC), National Institute of Health and Care Excellence (NICE), UK Caldicott Guardian Council, and Department of Health and Social Care (DH) as key bodies. Our documents included reports, guidelines, position papers, recommendations, and codes of practices published within the past 20 years. We checked each authoring body's website to ensure the document was the latest version. From our initial search results, we used backwards citation searching to identify additional relevant documents from other national bodies in England (from the Social Care Institute for Excellence (SCIE), Adass, and Local Government Association). We excluded speciality-specific documents (e.g., from Royal Colleges) to give our review boundaries.

Was this a rapid review? A desk review?

Yes, it was a desk review which we now state.

Might the authors provide a flow diagram showing how many potentially eligible results were screened?

As this was not a systematic review, and since we were following the process laid out by Dalglish et al., we did not document number of results and numbers screened for eligibility. However, we have

now added more detail on what we did within each step of the READ approach—please see the paper from the heading *Ready your materials* onwards.

Can the authors list all the professional bodies whose websites were searched?
Please see above.

What was done in cases where the websites did not yield results (e.g., required password access or had poor search engine functionality)? Were HCPs consulted?

We did not encounter this problem as the key bodies whose websites we searched yielded results. We consulted HCPs in deciding upon these key bodies, as we now explain (see paragraph above).

Please provide references for the methods used, such as the framework approach.
We have now added a reference for the Framework Method.

The sentence "questions such as whether... facilitated data analysis" is vague. In what way did they facilitate data analysis? Were they rows in the framework matrix? Some other way?

We have now changed this sentence to read: *As we read, we also made notes about whether documents referenced each other, what they recommended on the same issues, whether recommendations had changed over time, and the laws that underpinned the guidance. We also identified the ethical constructs underpinning the recommendations in guidance documents.*

The methods read like a qualitative study but if so, they would benefit from a statement on positionality (were analysers clinicians themselves?), reflexivity and other aspects of qualitative research practice.

We have now added

SD, a research fellow and applied social scientist, led on the analysis with support from GF, an academic GP. Having two different perspectives (clinical and non-clinical) was helpful for analysis and differences in interpretation of documents was resolved through discussion.

Can the wider project be named? Can a reference to the study website be provided or a protocol paper or a study registry link?

Our aim now states:

To improve practice, we led a project called Recording and Sharing Information about DVA in Healthcare (R&SDIH). It involved multiple research strands including an in-depth analysis of existing guidance about medical confidentiality, information sharing, and DVA for HCPs (and social care professionals). We present this analysis here. We aimed to highlight commonalities, inconsistencies, gaps, and ambiguities in guidance and in turn the tensions, dilemmas, and complex ethical issues with which HCPs are faced when caring for patients affected by DVA. The findings from this analysis, as well as findings from the other research strands, informed recommendations for good practice in England. R&SDIH was in turn part of a larger project called Health Pathfinder, which aimed to enhance the healthcare response to DVA. Reports from R&SDIH[33] and Health Pathfinder are published elsewhere [34].

FINDINGS

This section reads as though the authors are presenting themes from qualitative synthesis of the texts analysed but this terminology is not applied. It might harmonise the paper if consistent terminology (either uniform discussion of methods and themes or another set of terms more suited to the READ method?) were used across the methods and findings.

We have now made clear that the READ approach is indeed a form of qualitative analysis. But we understand that the term 'theme' might be confusing and imprecise as it suggests we used an inductive thematic analysis, which we did not. We have changed the word 'theme' and instead stated "*We present our findings under these headings below.*"

Table 3: please define GDPR when it is first used.

We have done this.

The early paragraphs could be more concise by removing the lengthy direct quotations, which are in any case repeated in Table 4.

Thank you for this suggestion but we feel it is clearer to leave the quotations in text. The quotations in Table 4 are different to those in the text.

p15 please specify and reference which code of practice is referred to here.

We had made an error in reference and cited the same code of practice twice (reference 46 and 52) – we have now corrected this and made sure each mention of 'code of practice' is referenced correctly (46).

p17 The sentence beginning "perpetrators' information..." is unclear. Are the authors stating this as a fact or paraphrasing what this document says?

Paraphrasing—thanks for pointing out that this sentence was unclear. We have now said 'The documents indicate that perpetrators' information...'

p19 The sentence beginning "the DH moreover clarifies" needs a reference, given there are multiple DH references.

Apologies this was a typo -we have now added the reference.

DISCUSSION

p19 Perhaps it is not true to say HCPs are faced with 13 documents, given that midwives are unlikely to read BMA guidance and vice versa?

We have changed this to *A striking initial finding was that 13 different documents all aim to guide good practice.*

p20 and p21 the direct quotes in the Discussion could be phrased in the authors' own words.

Thank you for this suggestion. We have reduced the first quote from Evan Stark, but we still use a shorter direct quotation because his work on coercive control has been ground-breaking and the way he explains it is eloquent and precise. We have rephrased the second quote to

Cohen and Caswell [55] criticise the 2017 GMC [39] guidance for not supporting confidentiality breaches in cases where victim-survivors (who are adult, and have no children under 18, so do not fall under safeguarding duties) are facing high-risk abuse but decline information-sharing or referral.

p24 I don't clearly follow the meaning of this final paragraph beginning "the aim of the research presented here..."

This was a convoluted way of saying 'we know it is counter-intuitive that we have conducted a project (R&SDIH) whose aim was to create guidelines when (a) there are already too many guidelines and (b) we do not even know if HCPs read and use guidelines. We have now deleted the paragraph and have added a sentence near the start of the paragraph to state:

Too many different guidance documents exist, and HCPs arguably need one set of recommendations to implement good practice. However,....[and here is where we explain that we don't even know if HCPs read and use guidelines].

CONCLUSIONS

Please state the countries involved for international readers, rather than saying "national".
We have gone through the paper and clarified (national bodies from England)

The summary of other work not presented in this paper does not belong under the conclusions of this paper.

Thank you, we have rewritten the conclusion and deleted most of what we said about the wider project, reducing it to *There is a need for coherent recommendations for cross-speciality clinical practice: in our wider project, an expert advisory group developed such a set of recommendations for all HCPs [66]*

The contents of Table 5 do not follow from the results presented. Perhaps they are more helpful as a supplementary resource?

We decided to delete Table 5

The conclusion that sharing without consent be done in a maternalistic way does not follow from the study presented. Perhaps it belongs in the other study?

We have removed it from the conclusion.

We have retained our reflection on the concept of maternalism in the Discussion because a key finding is that guidelines indicate a default stance of relying on consent and maintaining confidentiality, which in some cases of DVA, can be lethal. We think Sullivan's analysis is helpful because it shows HCPs that there is more nuance to the question of whether to be paternalistic (sharing without consent) vs non-paternalistic (i.e., relying on the patient's consent). This analysis is not used in the larger R&SDIH project, only in this paper.

Thank you for your helpful comments.

VERSION 2 – REVIEW

REVIEWER	Patricia Easteal University of Canberra
REVIEW RETURNED	17-Mar-2022

GENERAL COMMENTS	Thank you for adding to the Conclusion/Recommendations
--

REVIEWER	Roxanne Keynejad King's College London, Institute of Psychiatry
REVIEW RETURNED	27-Mar-2022

GENERAL COMMENTS	Thank you to the authors for comprehensively addressing my feedback and clearly outlining how they have done so in their response letter. I have no further suggestions for this resubmission.
--